

# Immunoglobulin superfamily member 10 is a novel prognostic biomarker for breast cancer

Mengxue Wang[1], Meng Dai[2], Yu-shen Wu[1], Ziying Yi[1], Yunhai Li[3] and Guosheng Ren[1]

[1] Chongqing Key Laboratory of Molecular Oncology and Epigenetics, The First Affiliated Hospital of Chongqing Medical University, Chongqing, China
[2] Department of Oncology, The First People's Hospital of Neijiang, Neijiang, Sichuan, China
[3] Department of Endocrine and Breast Surgery, The First Affiliated Hospital of Chongqing Medical University, Chongqing, China

Corresponding authors
Yunhai Li,
leeyh90@hospital.cqmu.edu.cn
Guosheng Ren, rengs726@126.com

## ABSTRACT

**Background:** Immunoglobulin superfamily member 10 (IGSF10) is a member of the immunoglobulin superfamily that is expressed at high levels in both the gallbladder and ovary. Currently, the role and possible mechanism of IGSF10 in breast cancer remain unclear.

**Method:** By applying real-time quantitative polymerase chain reaction (qRT-PCR) and immunohistochemistry (IHC), the expression of IGSF10 in breast cancer cells and tissues was detected. We collected the clinical information from 700 patients with breast cancer in The Cancer Genome Atlas (TCGA), and analyzed the relationship between IGSF10 expression and the clinicopathological features and survival outcomes of these patients. The potential mechanisms and pathways associated with IGSF10 in breast cancer were explored by performing a gene set enrichment analysis (GSEA).

**Results:** According to TCGA data, qRT-PCR and IHC experiments, levels of the IGSF10 mRNA and protein were significantly decreased in breast cancer tissues. IGSF10 expression was significantly correlated with age, tumor size, and tumor stage. Moreover, shorter overall survival (OS) and relapse-free survival (RFS) correlated with lower IGSF10 expression, according to the survival analysis. The multivariate analysis identified that IGSF10 as an independent prognostic factor for the OS (hazard ratio (HR) = 1.793, 95% confidence interval (CI) [1.141–2.815], $P = 0.011$) and RFS (HR = 2.298, 95% CI [1.317–4.010], $P = 0.003$) of patients with breast cancer. Based on the GSEA, IGSF10 was involved in DNA repair, cell cycle, and glycolysis. IGSF10 was also associated with the PI3K/Akt/mTOR and mTORC1 signaling pathways.

**Conclusions:** This study revealed a clear relationship between IGSF10 expression and the tumorigenesis of breast cancer for the first time. Therefore, further studies are needed to understand the mechanism of IGSF10 in breast cancer.

## INTRODUCTION

Breast cancer is a common malignancy that seriously threatens women's health. Approximately 2.1 million female patients were newly diagnosed with breast cancer worldwide in 2018. Breast cancer accounts for one-quarter of all female cancer cases (*Bray et al., 2018*). As a heterogeneous disease, the initiation and development of breast cancer are affected by both genetic and environmental factors (*Yang et al., 2019*). Despite continuous advances in surgical techniques, biological drugs and targeted therapies, breast cancer remains an arduous clinical problem (*Woolston, 2015*). Therefore, the identification of breast cancer biomarkers is crucial for obtaining an understanding of the tumorigenesis and accurate cancer prognosis, as biomarkers may assist with the clinical diagnosis and serve as potential tumor therapeutic targets in patients with breast cancer (*Costa-Pinheiro et al., 2015*; *Prensner et al., 2012*; *Qiao et al., 2019*).

Immunoglobulin superfamily member 10 (IGSF10) is a gene involved in cell differentiation and developmental processes (*Thutkawkorapin et al., 2016*). Mutations in IGSF10 delay human puberty (*Howard, 2018*; *Howard et al., 2016*). Moreover, during embryonic development, mutations in IGSF10 lead to the dysregulation of gonadotropin-releasing hormone (GnRH)-associated neuronal migration. Based on accumulating evidence, IGSF10 deficiency may lead to a transient GnRH deficiency and reversible congenital hypogonadotropic hypogonadism (*Amato et al., 2019*; *Howard, 2018*). Moreover, mutations in IGSF10 likely contribute to an increased risk of rectal and gastric cancers (*Thutkawkorapin et al., 2016*). As shown in the study by *Daino et al. (2009)* IGSF10 is significantly downregulated in a rat model of alpha-radiation-induced osteosarcoma. The expression of IGSF10 is downregulated in lung cancer tissues, and decreased expression of IGSF10 correlated with a poor prognosis for patients with lung cancer (*Ling et al., 2020*). However, the biological roles of IGSF10 in the majority of cancers have not been investigated, and its role in breast cancer remains largely unknown.

In the present study, the expression of IGSF10 in collected breast cancer tissues was examined using qRT-PCR and IHC. The clinicopathological features of the disease based on IGSF10 expression and Kaplan–Meier survival curves were analyzed using public data from The Cancer Genome Atlas (TCGA) database. In addition, a gene set enrichment analysis (GSEA) was performed to explore the potential mechanisms and signaling pathways by which IGSF10 may mediate breast tumorigenesis.

## MATERIALS AND METHODS

### Cell culture

The breast cancer cell lines: MDA-MB-231, MCF-7, BT-549, ZR-75-30, SKBR-3, and T47D (ATCC, Manassas, VA, USA) were maintained as previously described (*Zhang et al., 2019*). The normal mammary epithelial cell line MCF-10A was also maintained as previously described (*Debnath, Muthuswamy & Brugge, 2003*). All cell lines were cultured in a humidified incubator at 37 °C with an atmosphere containing 5% $CO_2$.

## Patients with breast cancer and tissue samples

TCGA data were utilized as previously described (Qiu et al., 2018). In the present study, we analyzed IGSF10 expression in 1,095 patients with breast cancer in TCGA database. We included 700 patients with breast cancer who had complete RNA-seq data and complete clinical information to analyze the clinical correlation between IGSF10 expression and breast cancer. The following clinical information was collected: age, tumor size, lymph node status, tumor, node, metastasis (TNM) stage, estrogen receptor (ER) status, progesterone receptor (PR) status, human epidermal growth factor receptor 2 (HER2) status, and follow-up information.

Breast cancer tissue samples were collected as previously described (Li et al., 2018). Specifically, we collected 52 pairs of breast tumor and adjacent normal tissues from patients with breast cancer during surgery between 2014 and 2016 at The First Affiliated Hospital of Chongqing Medical University. The collected tissues were used for real-time quantitative polymerase chain reaction (RT-qPCR) and immunohistochemistry (IHC) analyses. All specimens were stored in liquid nitrogen. The collection and use of the tissues were approved by the Institutional Ethics Committees of the First Affiliated Hospital of Chongqing Medical University. The approval number allocated to this study by the Institutional Ethics Committees is 2017 Research Ethics (2017-012).

## RNA isolation and RT-qPCR

As described in a previous study (Qiu et al., 2018), we followed the manufacturer's instructions and extracted the total RNA using TRIzol reagent (Life Technologies Inc., Gaithersburg, MD, USA). RT-qPCR of 21 paired tissues was performed with an ABI 7500 Real-Time PCR System (Applied Biosystems, Foster City, CA, USA) to examine IGSF10 expression. Relative quantification of the expression of the IGSF10 mRNA was standardized to the expression levels of GAPDH. The following primer pairs were used in the present study:

Forward primer (IGSF10): 5′-TTGGAGTTTGCCTGATGGAAC-3′;
Reverse primer (IGSF10): 5′-CGCTACCCCAACTTTGTTGAAG-3′;
Forward primer (GAPDH): 5′-GGAGCGAGATCCCTCCAAAAT-3′;
Reverse primer (GAPDH): 5′-GGCTGTTGTCATACTTCTCATGG-3′.

## IHC

The procedure used for IHC was described in a previous study (Li et al., 2018). An anti-IGSF10 rabbit polyclonal antibody (ab197671, 1:100, Abcam), a secondary antibody (ZSGB 1:100 SPN9001) and HRP (ZSGB 1:100 SPN9001) were used. Thirty-one paired tissues were subjected to IHC. The IHC staining intensity scoring criteria were as follows: 0, none; 1, weak; 2, medium; and 3, strong. The scoring criteria for the proportion of positive tumor cells were as follows: 0, <5%; 1, 5%–25%; 2, 26%–50%; 3, 51%–75%; and 4, >75%. An overall score was derived by multiplying the intensity and percentage scores.

## Bioinformatics analyses

The expression of IGSF10 in different subtypes of breast cancer was analyzed using UALCAN, a web portal for evaluating gene expression in different tumor subtypes stratified according to the various clinicopathological features of patients in TCGA database (*Chandrashekar et al., 2017*).

The expression of the IGSF10 mRNA in different breast cancer datasets was evaluated using Oncomine gene expression array datasets (*Rhodes et al., 2004*). The cutoff *P*-value and absolute fold change were defined as 0.01 and 2, respectively.

The relationship between IGSF10 expression and the prognosis of patients with breast cancer presenting different molecular subtypes was analyzed using a Kaplan–Meier plotter (http://kmplot.com/analysis/) (*Lánczky et al., 2016*). The Affymetrix probe set ID of IGSF10 is 230670_at. Patients were automatically stratified into *IGSF10*-high and *IGSF10*-low groups according to the mean expression of the IGSF10 mRNA.

## GSEA

This method was described in previous study (*Jiao et al., 2018*). We performed a GSEA (http://software.broadinstitute.org/gsea) to explore the association between *IGSF10* expression and biological processes/pathways according to the instructions of the user guide. We performed the GSEA using a microarray dataset (GSE1456) and TCGA microarray dataset.

## Additional statistical analyses

All statistical analyses were performed using SPSS software (version 23.0). OS and RFS were calculated by constructing Kaplan–Meier curves. The differences between two groups were evaluated using Student's t test. Significance was set to a *P*-value less than 0.05.

# RESULTS

## The expression of IGSF10 in breast cancer and its clinicopathological features

We examined the expression of the IGSF10 mRNA in 1095 patients with breast cancer in TCGA database. Based on our results, the IGSF10 mRNA was expressed at higher levels in adjacent normal tissues than in breast cancer tissues (Fig. 1A). We then detected the differences in IGSF10 expression in 21 paired tissue samples using RT-qPCR. Consistent with the results from TCGA database, IGSF10 expression was substantially downregulated in breast cancer tissues (Fig. 1B; Table S1). We collected 31 pairs of breast cancer and corresponding normal tissues and performed IHC. The staining scores of the breast cancer tissues were significantly lower than the adjacent normal tissues (Figs. 1C–1G). Finally, we examined the expression of the IGSF10 mRNA in breast cell lines. IGSF10 was expressed at higher levels in the normal breast epithelial cell line MCF10A than in the breast cancer cell lines (Fig. 1E; Table S2).

Seven hundred patients with breast cancer in the TCGA cohort were analyzed to further confirm the correlation between IGSF10 expression and breast cancer (Table S3).

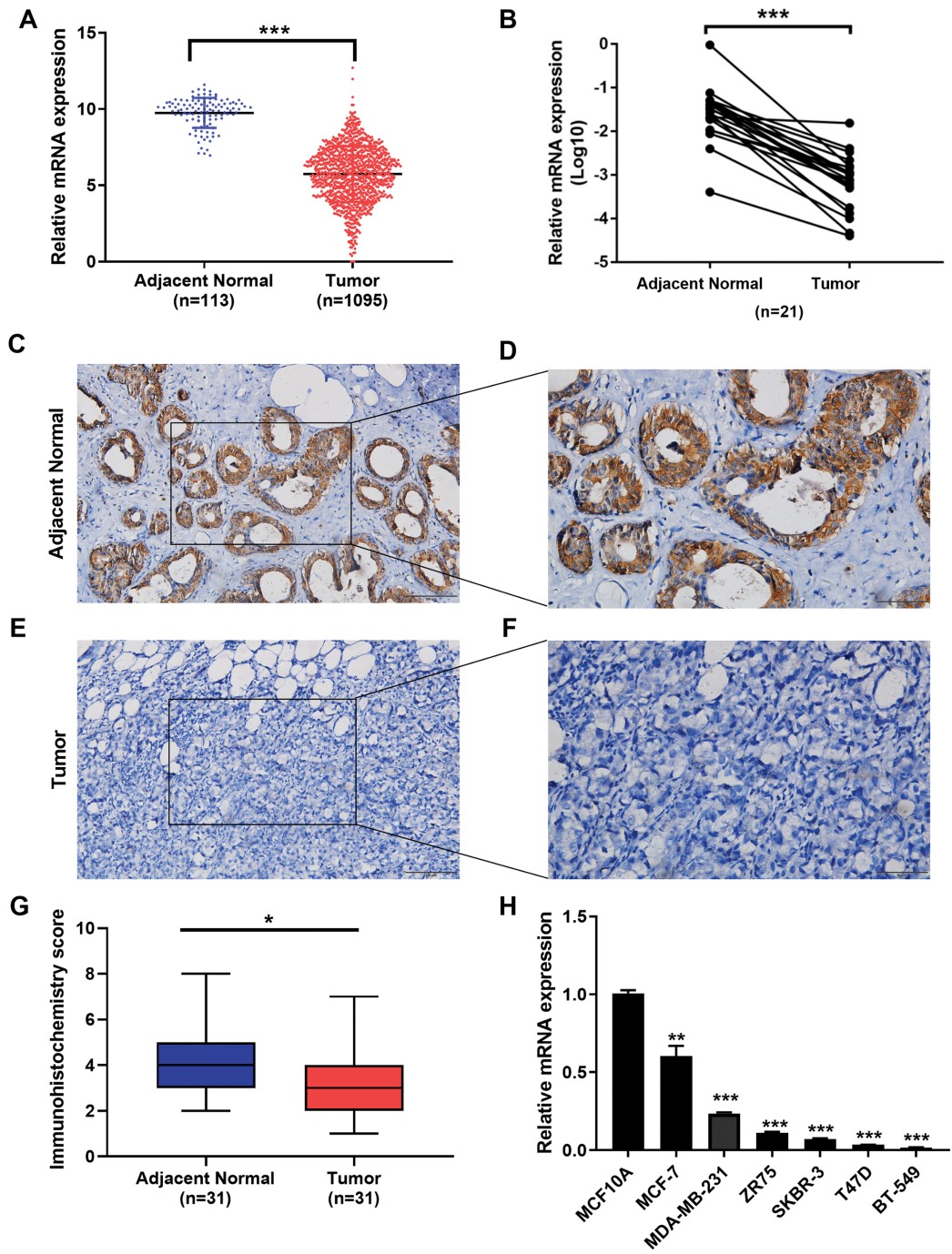

**Figure 1 Expression of IGSF10 in breast cancer.** (A) Bioinformatics analysis of IGSF10 expression in TCGA database. (B) The expression of the IGSF10 mRNA in BC tissues and matched adjacent normal tissues was evaluated using qRT-PCR (*n* = 21). (C–F) Representative images of IHC staining in BC specimens and adjacent normal breast tissues. (G) IHC score for the level of the IGSF10 protein expression in 31 BC tissues (IHC score: 3.12 ± 2.04) and 31 normal tissues (IHC score: 4.45 ± 2.13). Data are presented as mean ± SD, unpaired *t*-test, *P < 0.05. (H) qRT-PCR was used to examine IGSF10 expression in human breast cancer cells and MCF-10A cells; *P < 0.05, **P < 0.01, and ***P < 0.001.

Table 1 **Clinical correlation of *IGSF10* in breast cancer.** Seven hundred patients with breast cancer in the TCGA cohort were analyzed to further confirm the correlation between IGSF10 expression and breast cancer.

| Characteristic | Number of cases | IGSF10 | | |
|---|---|---|---|---|
| | | High (n) | Low (n) | P-value |
| Age | | | | |
| <50 | 193 | 125 | 68 | <0.001* |
| ≥50 | 507 | 242 | 265 | |
| Tumor size | | | | |
| T1 | 183 | 114 | 69 | 0.003* |
| T2 | 418 | 207 | 211 | |
| T3 | 75 | 39 | 36 | |
| T4 | 24 | 7 | 17 | |
| Lymph node metastasis | | | | |
| N0 | 342 | 181 | 161 | 0.865 |
| N1 | 236 | 119 | 117 | |
| N2 | 85 | 47 | 38 | |
| N3 | 37 | 20 | 17 | |
| TMN stage | | | | |
| I | 124 | 78 | 46 | 0.03* |
| II | 407 | 203 | 204 | |
| III | 156 | 82 | 74 | |
| IV | 13 | 4 | 9 | |
| ER | | | | |
| Positive | 539 | 284 | 255 | 0.800 |
| Negative | 161 | 83 | 78 | |
| PR | | | | |
| Positive | 473 | 249 | 224 | 0.870 |
| Negative | 227 | 118 | 109 | |
| HER-2 | | | | |
| Positive | 102 | 53 | 49 | 0.918 |
| Negative | 598 | 314 | 284 | |
| Triple negative breast cancer | | | | |
| Yes | 119 | 61 | 58 | 0.779 |
| No | 581 | 306 | 275 | |

Notes:
ER, estrogen receptor; PR, progesterone receptor.
* $p < 0.05$ was considered statistically significant.

IGSF10 expression correlated with age ($P < 0.001$), tumor size ($P = 0.003$), and TNM stage ($P = 0.03$) (Table 1).

## High IGSF10 expression correlated with a better prognosis for patients with breast cancer

The associations of IGSF10 expression with overall survival (OS) and relapse-free survival (RFS) were evaluated using Kaplan–Meier survival curves. Patients in TCGA dataset were

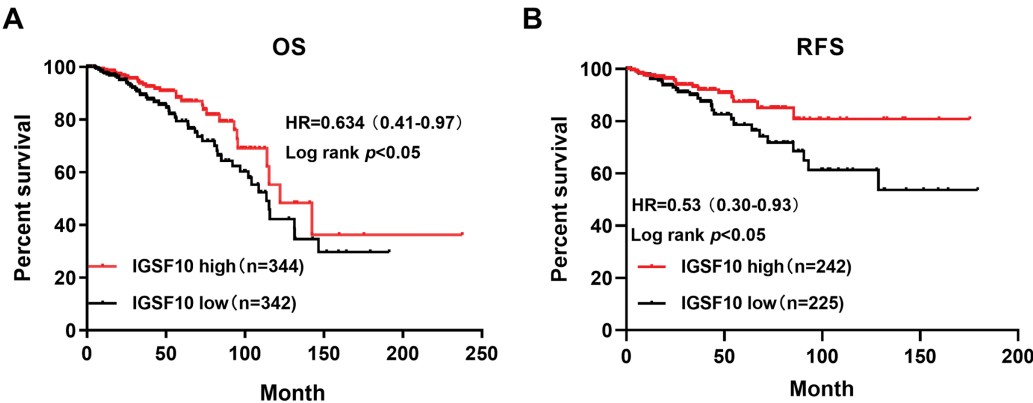

**Figure 2 Kaplan–Meier survival curve was plotted with TCGA cohort by stratifying patients into IGSF10 high and low groups with median expression value.** Kaplan–Meier survival curve of TCGA patients with breast cancer stratified into the IGSF10-high and IGSF10-low groups based on the median expression level. $P < 0.05$ was considered a statistically significant. (A) Curves showing the OS of patients with breast cancer. (B) Curves showing the RFS of patients with breast cancer.

stratified by the median IGSF10 mRNA expression level (Table S4). Patients with high IGSF10 expression were significantly more likely to experience prolonged OS (hazard ratio (HR) = 0.63, 95% confidence interval (CI) [0.41–0.97], $P < 0.05$) (Fig. 2A) and RFS (HR = 0.53, 95% CI [0.30–0.93], $P < 0.05$) (Fig. 2B) than patients with low IGSF10 expression. Subsequently, we used the UALCAN database to further evaluate the prognostic value of IGSF10 by stratifying patients into different molecular subtypes. Decreased levels of the IGSF10 mRNA were observed in luminal, HER2-positive, and triple-negative breast cancer samples compared with normal samples (Fig. 3A). Low IGSF10 expression was significantly correlated with a shorter OS of patients with basal (HR = 0.44, 95% CI [0.22–0.86], $P = 0.013$), luminal A (HR = 0.47, 95% CI [0.25–0.88], $P = 0.017$), and HER2+ (HR = 0.28, 95% CI [0.09–0.81], $P = 0.012$) breast cancer subtypes (Figs. 3B–3E). However, a significant relationship was not observed between the expression of IGSF10 and OS of patients with the luminal B subtype (HR = 0.61, 95% CI [0.3–1.23], $P = 0.17$) (Fig. 3D). The multivariate Cox regression analysis of TCGA patients with breast cancer showed that IGSF10 was an independent prognostic factor for OS (HR = 1.793, 95% CI [1.141–2.815], $P = 0.011$) and RFS (HR = 2.298, 95% CI [1.317–4.010], $P = 0.003$) (Table 2).

## Potential biological roles and signaling pathways related to IGSF10

Potential mechanisms and signaling pathways that may be related to the ability of IGSF10 to regulate the development of breast cancer were explored by conducting a GSEA. According to the median value of IGSF10 expression in the microarray dataset (GSE1456) and TCGA dataset, we assigned patients to two groups. Nine gene sets were enriched in the GSE1456 dataset and 16 gene sets were enriched in TCGA dataset ($P < 0.05$; false discovery rate (FDR) < 0.25) (Figs. 4A and 4B; Table S5). Interestingly, IGSF10 expression was positively correlated with several cancer-related biological processes, including DNA repair (HALLMARK_DNA_REPAIR), cell cycle (HALLMARK_G2M_CKECKPOINT),

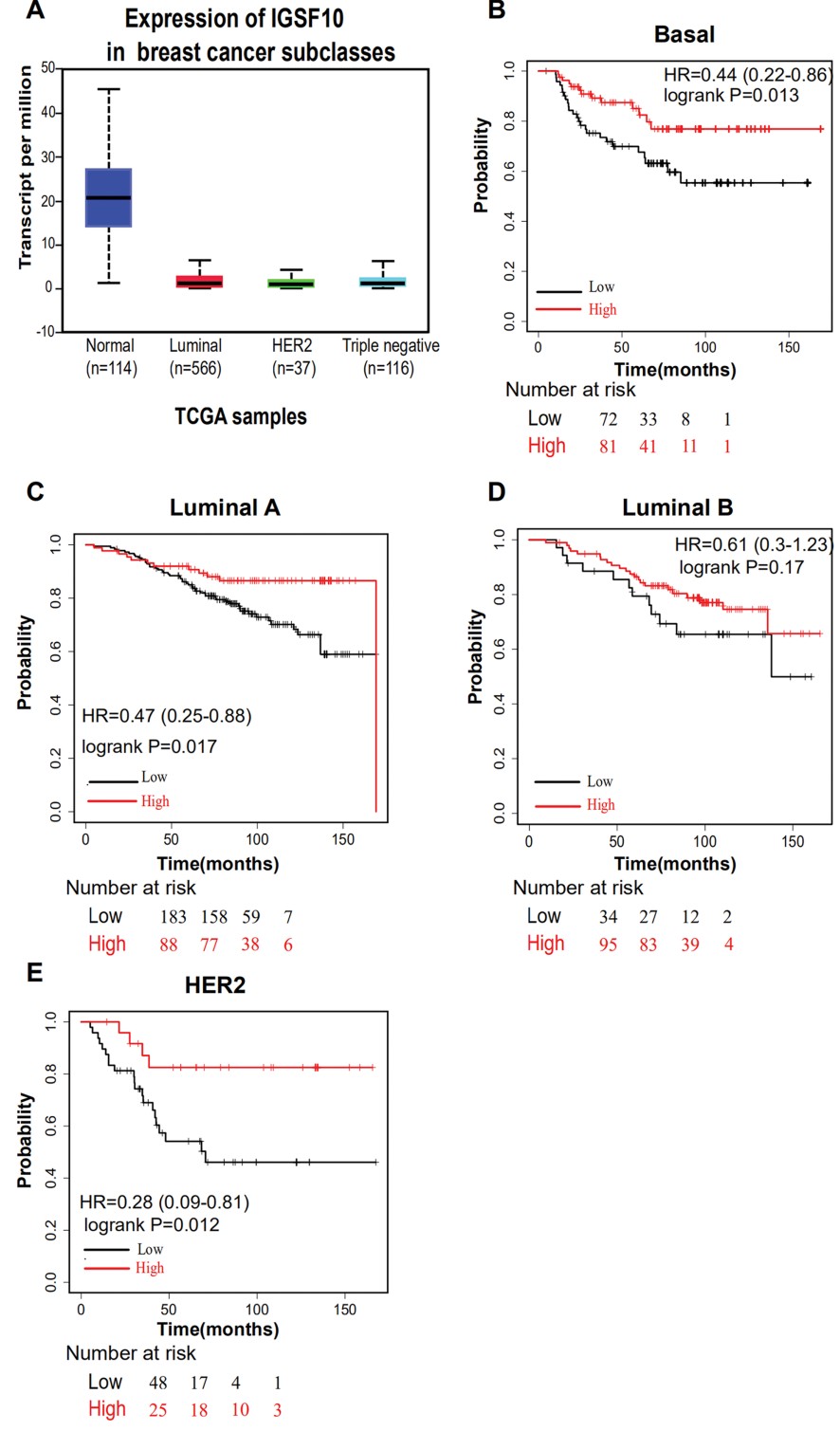

**Figure 3 Overall survival of different molecular subtypes of breast cancer.** (A) IGSF10 expression in patients with different molecular subtypes of breast cancer in TCGA database. (B) Basal breast cancer, (C) luminal A breast cancer, (D) luminal B breast cancer, and (E) HER2+ breast cancer. All the curves showing OS were plotted using the Kaplan–Meier plotter (http://kmplot.com/analysis/). $**P < 0.01$.

**Table 2 Univariate and multivariate Cox regression analysis of IGSF10 in the TCGA cohort.**

| Variants | OS | | | | | | RFS | | | | | |
| --- | --- | --- | --- | --- | --- | --- | --- | --- | --- | --- | --- | --- |
| | Univariate analysis | | | Multivariate analysis | | | Univariate analysis | | | Multivariate analysis | | |
| | HR | 95% CI | P-value | HR | 95% CI | P-value | HR | 95% CI | P-value | HR | 95% CI | P-value |
| Age (<50 vs. ≥50) | 0.597 | [0.358–0.997] | 0.049* | 0.626 | [0.367–1.069] | 0.086 | 0.768 | [0.433–0.945] | 0.041* | 0.669 | [0.373–1.245] | 0.178 |
| Tumor size (T1/T2 vs. T3/T4) | 0.825 | [0.493–1.380] | 0.464 | | | | 0.614 | [0.322–1.170] | 0.138 | | | |
| Lymph node (N0 vs. N1/N2/N3) | 0.603 | [0.384–0.947] | 0.028* | 0.954 | [0.534–1.704] | 0.873 | 0.753 | [0.443–1.279] | 0.294 | | | |
| TNM stage (I/II vs. III/IV) | 0.482 | [0.311–0.747] | 0.001* | 0.538 | [0.307–0.944] | 0.031* | 0.467 | [0.359–0.785] | 0.001* | 0.597 | [0.347–0.842] | 0.012* |
| ER (negative vs. positive) | 1.197 | [0.734–1.951] | 0.471 | | | | 1.056 | [0.584–1.909] | 0.858 | | | |
| PR (negative vs. positive) | 1.489 | [0.960–2.311] | 0.076 | | | | 0.958 | [0.547–1.680] | 0.882 | | | |
| HER2 (negative vs. positive) | 1.093 | [0.563–2.122] | 0.793 | | | | 1.388 | [0.626–3.077] | 0.419 | | | |
| IGSF10 (low vs. high) | 1.645 | [1.054–2.569] | 0.029* | 1.793 | [1.141–2.815] | 0.011* | 2.102 | [1.222–3.615] | 0.006* | 2.298 | [1.317–4.010] | 0.003* |

Notes:
OS: overall survival; RFS: relapse-free survival; HR: hazard ratio; CI: confidence interval.
* $P < 0.05$ was considered statistically significant.

and glycolysis (HALLMARK_GLYCOLYSIS) pathways in both datasets (Figs. 4C–4E). The PI3K/Akt/mTOR and mTORC1 signaling pathways were also associated with IGSF10 (Figs. 4F–4G). Moreover, in TCGA dataset, the transforming growth factor-β (TGF-β) signaling pathway (HALLMARK_TGF_BETA_SIGNALING), epithelial mesenchymal transition (EMT) (HALLMARK_EPITHELIAL_MESENCHYMAL_TRANSITION) and tumor necrosis factor (TNF) signaling pathway (HALLMARK_TNFA_SIGNALING_VIA_NFKB) were significantly enriched in the IGSF10-low group (Fig. S1A). These results indicated a possible mechanism underlying the role of IGSF10 in the tumorigenesis of breast cancer.

## DISCUSSION

In recent years, numerous molecular prognostic biomarkers have been identified and validated in cancers, including breast cancer (Nicolini, Ferrari & Duffy, 2018). In the present study, we identified IGSF10 as a potential prognostic biomarker for breast cancer and described a possible mechanism underlying its role in the tumorigenesis of breast cancer.

In the present study, we explored the role of IGSF10 in breast cancer by analyzing TCGA data and performing RT-qPCR and IHC. Our data indicated that IGSF10 expression was significantly downregulated in breast cancer tissues. Consistent with our results, multiple datasets in the Oncomine database suggested that IGSF10 expression was down-regulated in breast cancer tissues (absolute fold change > 2) including TCGA Breast, Karnoub Breast (Karnoub et al., 2007), Zhao Breast (Zhao et al., 2004), Richardson Breast 2 (Richardson et al., 2006), and Finak Breast (Finak et al., 2008) datasets (Table S6). Based on the analysis of the data in the UALCAN database, we found that IGSF10 expression correlated with the molecular subtype of breast cancer. In addition, IGSF10

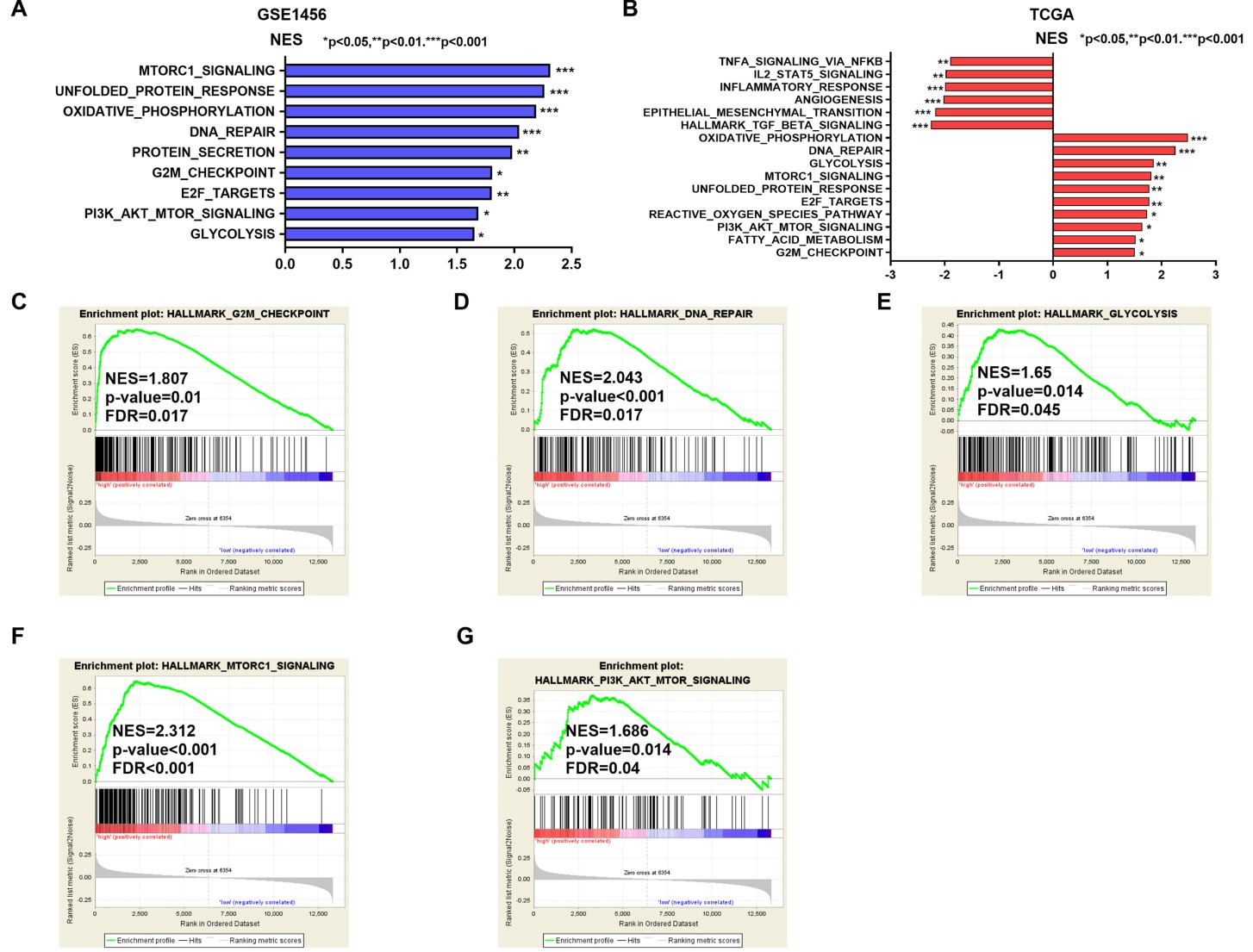

**Figure 4 The gene sets that were significantly associated with *IGSF10* with normal *P*-value < 0.05 and false discovery rate (FDR) < 0.25.** Gene sets with a normal *P*-value < 0.05 and an FDR < 0.25 were considered significant. Gene sets were ranked by the normalized enrichment score (NES). (A) Gene sets enriched in the GSE1456 dataset. (B) Gene sets enriched in the TCGA dataset. (C–E) GSEA enrichment plot showing that IGSF10 expression was positively associated with DNA repair, cell cycle, and glycolysis. (F and G) GSEA enrichment plot showing that IGSF10 expression was positively correlated with the PI3K/Akt/mTOR and mTORC1 signaling pathways.

expression was closely associated with age, tumor size, and TNM stage. Accordingly, IGSF10 may play a crucial role in breast cancer and have the potential to be targeted by anticancer therapy. Moreover, the survival analysis indicated that patients with breast cancer presenting higher IGSF10 expression experienced prolonged OS and RFS. The multivariate analysis identified IGSF10 as an independent prognostic factor for patients with breast cancer. Interestingly, in the subgroup analysis, IGSF10 expression was significantly correlated with OS in patients with basal, luminal A and HER2-positive breast cancer. Thus, IGSF10 may be a prognostic biomarker for breast cancer.

IGSF10 may exert an important effect on tumorigenesis. Ling and colleagues claimed that IGSF10 knockout promotes the development of lung cancer and that IGSF10 mainly activates the integrin-β1/FAK pathway in lung cancer (*Ling et al., 2020*). In one family with gastric and colorectal cancers, *Thutkawkorapin et al. (2016)* identified 12 new nonsynonymous single nucleotide variants in 12 different genes, including IGSF10, with potential contributions to an increased cancer risk. *Chang et al. (2017)* identified new mutations in patients with endometrial cancer in Taiwan by performing whole-exome sequencing and identified a potential association between IGSF10, a passenger gene, with endometrial cancer. However, to our knowledge, no studies have reported the possible functions and mechanisms of IGSF10 in breast cancer.

During the past decade, accumulating evidence has revealed clear correlations between immunoglobulin superfamily members and human diseases. For instance, loss-of-function mutations in IGSF1 result in an X-linked syndrome of central hypothyroidism and testicular enlargement. IGSF1 mutations in male patients lead to a late increase in testosterone levels (*Howard et al., 2016*; *Roche et al., 2018*; *Sun et al., 2012*). Significantly prolonged OS was observed in pediatric patients with mixed-lineage leukemia-rearranged acute monoblastic leukemia with t(9; 11) (p22; q23) and high IGSF4 expression than in patients with low IGSF4 expression (*Kuipers et al., 2011*). As shown in the study by Wang et al., IGSF8 promotes melanoma proliferation and metastasis by negatively regulating the TGF-β signaling pathway (*Wang et al., 2015*).

In the present study, potential biological roles and signaling pathways that may be related to IGSF10 expression in breast cancer were analyzed by conducting a GSEA. Several biological processes, including DNA repair, the cell cycle, and glycolysis, were associated with IGSF10. Among these processes, the genomic integrity is maintained through DNA repair pathways. The dysregulation of DNA repair leads to changes in the genome and causes physiological changes in cells that drive tumor initiation (*Jeggo, Pearl & Carr, 2016*; *Khanna, 2015*; *Mouw et al., 2017*). The cell cycle regulates tumor growth and glycolysis modulates the heterogeneity of the tumor microenvironment. These biological processes are related to tumor progression, metastasis and drug resistance (*Jahagirdar et al., 2018*). Moreover, in human malignancies, the mTORC1 and PI3K/Akt/mTOR signaling pathways are usually abnormally activated and promote the development of malignancies (*Hare & Harvey, 2017*). According to previous studies, mTORC1 promotes cell growth by activating key anabolic processes and the dysregulation of mTORC1 is the basis of many human cancers (*Ben-Sahra & Manning, 2017*; *Keppler-Noreuil et al., 2016*). The PI3K/Akt/mTOR pathway is related to various biological processes in breast cancer, such as tumorigenesis, cellular transformation, tumor progression, and drug resistance (*Guerrero-Zotano, Mayer & Arteaga, 2016*). Therefore, we speculated that IGSF10 might mechanistically regulate the growth of breast cancer cells through the mTORC1 and PI3K/Akt/mTOR signaling pathways. Intriguingly, IGSF10 was associated with EMT, the TGF-β signaling pathway and the TNF signaling pathway in TCGA database. The TGF-β signaling pathway was reported to be associated with various tumors and it regulates the biological processes in multiple cancers, including growth, migration, invasion, apoptosis and the EMT (*Bedi et al., 2012*; *Tang et al., 2017*; *Yu et al., 2018*; *Zhao et al., 2018*). The EMT

plays crucial roles in the metastasis and invasion of breast cancer by regulating cell motility and invasiveness (*Feng et al., 2016*). Moreover, TNF-α is strongly correlated with inflammation in breast tumors, and an increase in its expression is strongly correlated with relapse and advanced disease (*Katanov et al., 2015*). However, further studies are needed to elucidate the role of IGSF10 in breast cancer and the detailed mechanisms by which IGSF10 modulates these related signaling pathways.

## CONCLUSIONS

In summary, IGSF10 was expressed at a low level in breast cancer. IGSF10 expression was significantly correlated with age, tumor size, and tumor stage. More importantly, IGSF10 was an independent prognostic factor for better outcomes in patients with breast cancer. In addition, the GSEA results identified significant associations between IGSF10 expression and DNA repair, cell cycle, glycolysis, and the mTORC1 and PI3K/Akt/mTOR signaling pathways. Overall, we suggested a novel role for IGSF10 in breast cancer. Our data may provide new insights into the identification of potential therapeutic targets in patients with breast cancer.

## ACKNOWLEDGEMENTS

I greatly appreciate the assistance and encouragement from my tutor.

### Funding

This study was supported by the National Natural Science Foundation of China (Nos. 81472475 and 81102007), the Chongqing Science & Technology Commission (No. cstc2016jcyjA0313) and the Scientific Research Foundation of Chongqing Medical University (No. 201408). The funders had no role in study design, data collection and analysis, decision to publish, or preparation of the manuscript.

### Grant Disclosures

The following grant information was disclosed by the authors:
National Natural Science Foundation of China: 81472475 and 81102007.
Chongqing Science & Technology Commission: cstc2016jcyjA0313.
Scientific Research Foundation of Chongqing Medical University: 201408.

### Competing Interests

The authors declare that they have no competing interests.

### Author Contributions

- Mengxue Wang conceived and designed the experiments, performed the experiments, analyzed the data, prepared figures and/or tables, authored or reviewed drafts of the paper, and approved the final draft.
- Meng Dai performed the experiments, prepared figures and/or tables, and approved the final draft.

- Yu-shen Wu performed the experiments, prepared figures and/or tables, and approved the final draft.
- Ziying Yi performed the experiments, prepared figures and/or tables, and approved the final draft.
- Yunhai Li conceived and designed the experiments, analyzed the data, prepared figures and/or tables, authored or reviewed drafts of the paper, and approved the final draft.
- Guosheng Ren conceived and designed the experiments, authored or reviewed drafts of the paper, and approved the final draft.

### Human Ethics

The following information was supplied relating to ethical approvals (i.e., approving body and any reference numbers):

This study was approved by the Institutional Ethics Committees of the First Affiliated Hospital of Chongqing Medical University (research ethics number 2017-012).

### Data Availability

KM PLOTTER is available at http://kmplot.com/analysis/.

Data is available at NCBI GEO: GSE1456.

Additional breast cancer data are available at TCGA: https://www.cancer.gov/types/breast.

Clinical and gene expression data (ID:230670) are available at the Cancer Genomics Browser of the University of California, Santa Cruz (UCSC) (https://genome.ucsc.edu/cgi-bin/hgGateway), version: 2015-02-24.

Experimental data are available as Supplemental Files.

### Supplemental Information

Supplemental information for this article can be found online at http://dx.doi.org/10.7717/peerj.10128#supplemental-information.

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
