# Peer review of "Immunoglobulin superfamily member 10 is a novel prognostic biomarker for breast cancer"

_PeerJ, doi:10.7717/peerj.10128_

## Round 0.1 · original submission · Major Revisions

Three specialists evaluated this submission. They all have concerns regarding this manuscript. In my view, this paper needs a major revision. One of the reviewers asked to send a copy in English of the ethical approval document and the informed consent form template submitted.

·

Basic reporting

The authors report that IGSF10 is a novel prognostic biomarker in breast cancer and that their study, for the first time, revealed a clear relationship between IGSF10 and the tumorigenesis of breast cancer.
It is an interesting article with quality microphotographs, well-supported discussion and has potential.
However, I found the presentation of the data very confusing in some areas and a much-revised version of the article would be required.
Specifically, I would like to address the following issues:

Major
- The results from the IGSF10 analyses on their cohort should be presented clearly on the results and discussed thoroughly.
- The findings on their cohort and the TCGA data should be presented on a separate paragraph, on the results section.
- Tables presenting their data concerning IGSF10 mRNA and protein expression, as well as the correlations with the clinicopathological data, would be a significant add-on for the study. It could be added as a Supplemental Tables.
- The authors should present the number of tumour and adjacent normal tissues from breast cancer patients that they used in this study on the methods. The number of the cases is 159 as mentioned on the results (section 3 - Potential biological roles and signalling pathways related to IGSF10)?
- The authors should add a Remark diagram

Minor
- Add the source of the secondary antibody and HRP used on the study.
- Figure 1, panel A, variable “Tumor”. The number of tumor cases (1102) is probably wrong!
- Figure 1, panel A, variable “Paratumor”. The authors here denote “normal tissue”. Should be corrected.
- Figure 2. Add the number of the patients included on the KM analysis
- The sentence “In Oncomine database, the mRNA expression of IGSF10 was much lower in breast cancer than normal tissues within datasets including TCGA Breast, Karnoub Breast (Karnoub et al., 2007), Zhao Breast (Zhao et al., 2004), Richardson Breast 2 (Richardson et al., 2006), and Finak Breast (Finak et al., 2008) (Table 1).”, should be transferred to the discussion and the table as Supplemental Table 1.
- The entitled on the results section “The prognostic value of IGSF10 in breast cancer”, refers not only to prognostic value of IGSF10, but and the correlation of the particular molecule with other clinicopathological variables. The title of the section should be changed accordingly.
- The Table 3 in which cohort refers? The TCGA study? If yes should be mentioned on the legend.
- Figure 1, panel D. The data should be presented as box- plot.
- Supplemental figures 1-18 should be deleted
- The “breast_cancer_tissue”, “CELL_LINE”, “GSEA” , “survival”, “TCGA clinical_data” should be added as Supplemental tables, but it must mentioned on the text.

Experimental design

The experimental design is good. Changes to the presentation of the data should be made.

Validity of the findings

Conclusions are well stated, statistically sound and controlled.

·

Basic reporting

Cell Culture: The authors have used 7 different cell-lines in this study. It is advisable that the authors provide cell-line authenticity certification to ensure that it is the right cell-lines as claimed, due to a lot of cell-line misidentification in recent times. Additionally, this also increases the transparency of the project.

Line 101 – 107: How many tissue samples were collected from patients?

Line 111 – Manufacturer’s instructions

Line 151 – 230670_at (Kindly use the correct identifiers)

Supplementary Oncomine figure shows that around 11 datasets were originally considered. Why were the excluded from the final study?

Experimental design

How were the adjacent normal samples confirmed as non-cancerous? Figure 1B shows some cases where the expression difference between adjacent and tumour are not very significant. Perhaps this is due to adjacent normal not being completely “tumour-free” in all aspects?

Validity of the findings

The expression data from the 21 patients where RT-PCR was performed has not been included in any tables. Perhaps the authors can shed more light on this – what was the distribution like in these 21 samples? How many were IDC/ ILC type? What was the ER/PR/HER2 classification? Though this is a small dataset when compared to TCGA, it will be interesting to see if it matches the data.

The results presented are indeed interesting. However, it would be too early to consider IGSF-10 as a biomarker with such a limited validation cohort. The authors can also check for associated genes which have correlated expression with IGSF-10 in the specified pathways from GSEA analysis.

The results in Table 2 from TCGA cohort indicate that the expression of IGSF 10 decreases with stage-wise progression of BC. Was the same observed in the 21 tissue samples processed for RT-PCR? If not, what are the authors' thoughts on this?

Additional comments

This manuscript by Wang et al. explores the role of IGSF-10 as a potential biomarker of breast cancer. Though the concept of the paper has been explained well, the manuscript would further benefit from being proofread by a native speaker or a language editor.

The authors have provided IHC images & some analysis data has supplementary files – but relevant Figure/Table legends have not been provided. It would be easier if the authors can amend this suitably.

·

Basic reporting

Please have a look at Minor concerns: 1 to 2.

Experimental design

Please have a look at Major concerns: 1 to 7.

Validity of the findings

No comment.

Additional comments

Breast cancer is one of the predominant types of tumors in women worldwide. Despite great progress in the early diagnosis and treatment recently, drug resistance and distal metastasis remain major causes of cancer-related mortality. Immunoglobulin superfamily member 10 (IGSF10), as a member of the immunoglobulin superfamily, is broadly expressed in both gall bladder and ovary. In this manuscript, Dr. Mengxue Wang and colleagues looked at the role of the IGSF10 in diagnosis and prognosis of breast cancer. The study’s conclusion showed that IGSF10 was expected to be used to predict the prognostic survival of breast cancer. Although the current study is interesting, there are some major concerns that need to be addressed before consideration of publication.

Major concerns:
1. “Materials & Methods – Patients and tissue samples of breast cancer” Line 105-106. Please provide the approval number of this study in the Institutional Ethics Committees.
2. “Materials & Methods – Patients and tissue samples of breast cancer” Line 102-104. Please provide the table of the clinical and pathological information for the “Tissue samples of breast cancer” as same as “fully clinical information of TCGA”. In addition, please analyze the correlation between IHC score and the clinicopathological information of breast cancer patients. (Line 124-139, and Line 188-206)
3. “Result – 1. The expression of IGSF10 in breast cancer” Line 172-174, Figure 1. “…the mRNA expression level of IGSF10 was significantly down-regulated in breast cancer tissues compared with normal tissues.” There is a difference between tumor adjacent normal tissue and normal tissue. Be sure to use well-defined words throughout the manuscript.
4. “Result – 1. The expression of IGSF10 in breast cancer” Line 180-183, Figure 1C and D. Please provide a larger magnification of the micrograph. From the available images in figure 1C, it appears that the normal breast tissue is closer to hyperplasia.
5. “Result – 1. The expression of IGSF10 in breast cancer” Line 183-186. Please provide the protein expression level of IGSF10 in cell line.
6. “Result – 2. The prognostic value of IGSF10 in breast cancer” Line 194-195. “Patients with high expression level of IGSF10 were significantly associated with better OS (Figure 2A) and RFS (Figure 2B) than those with low levels of IGSF10.” Please explain how to define the high and low expression groups of IGFS10. In the survival curve of figure 2, the case number of each group, Hazard Ratio, 95% confidence interval of ratio and P value should be indicated.
7. “Result – 3. Potential biological roles and signaling pathways related to IGSF10” Line 208-218. Why GSE1456 was used in GSEA analysis? TCGA dataset can also be used for GSEA analysis. Therefore, I suggest the authors should add the GSEA analysis of the TCGA dataset and display the common pathway appearing in the two datasets (TCGA and GEO1456), so as to increase readers' understanding of the potential functions of IGSF10.

Minor comments:
1. All images need to be reformatted to ensure the readability and aesthetics of the comments.
2. Be sure to use well-defined words throughout the manuscript.

---

## Round 0.2 · Major Revisions

As you can see, reviewers gave split decisions ranging from accept to reject. In my view, the manuscript requires another round of major revision to address the remaining issues raised by the reviewers #2 and #3.

·

Basic reporting

The manuscript in the current form is adequate for publication.

Experimental design

All my comments are answered.

Validity of the findings

No comment

Additional comments

In Figure 1, panel A, C, G, the word “Paratumor” should be corrected to "normal tissue". All other comments are adequately answered.

·

Basic reporting

Line 28: "Broadly expressed"? Do the authors mean high expression or widespread expression throughout the organ? The intended information is not clearly conveyed.

Line 62: It would be advisable to follow a uniform pattern of providing citations.

Line 64-65: Previous study/ previous studies? There are 2 citations

Line 88-89: The authors may add a note that MCF-10A is a non-tumorigenic cell-line. It would help the readers better associate with the research.

Line 93: the line may be re-phrased to specify "used/utilized" rather than collected, as there is no actual collection happening in this case. Additionally, a citation to the original Breast cancer TCGA paper would also be advisable.

Line 94: 700 samples were used in this study - were the other samples excluded for any specific reason? What were the inclusion/exclusion criteria?

Line 99-100: Was sample collection during the time of surgery, or during a biopsy? Information not provided. The number of samples collected also is not mentioned here.

Line 134: fold change of 2 or absolute fold change?

Line 145: How was GSE1456 chosen?

Line 154: Which is the right number of samples? 700 or 1095? Please clarify. There seems to be conflict between the methods and results sections.

Line 156: Figure number?

Figure 1A/B: The authors can either label it as adjacent normal or add a note in the text as para-tumor. It would be easy for the readers if uniform nomenclature is used throughout the manuscript.

Figure 1H: label to be "MDA-MB 231" - to be corrected.

Experimental design

The authors mention 21 samples (paired) for qPCR and 31 for IHC. How many samples were used for both? Was the data compared? If not, what was the reason why the same patient sample was not used, especially if collected from a surgery where there will be more tissue availability?

Line 157: How many times was the qPCR performed? Were the results consistent across repetitions? What are the values shown in Supplemental Table 1? Are they Ct values? In many cases, there seems to be no difference between the normal and tumor. There is no legend/key provided. Similar issue for S. Table 2 as well.

Validity of the findings

The findings show no or very low significance in most of the clinical parameters taken from the large TCGA dataset when compared between the high and low gene expression groups. Only in T1, there is a significance, but it may be due to a number of factors, and hard to convince as a biomarker for breast cancer.

The authors will have to validate their findings in a larger cohort, especially with an overlap of samples used in IHC and qPCR, to be more confident of the results.

Additional comments

The authors' responses to the previous review comments seemed convincing. Sufficient proof regarding the doubts on IRB approval, consent forms, and cell-line authenticity have been provided. However, the manuscript as such is still lacking some critical elements, as pointed out above. It would be in the best interest of the work, to address these issues at the earliest.

The authors mention in the rebuttal that the manuscript has been proof-read, however, there are still many factual and textual errors in the paper. The general flow in the manuscript also seems erratic. Results section 1 mentions TCGA data twice, but different numbers and causalities. Kindly look into it patiently and make sure there are no oversights.

·

Basic reporting

No comment.

Experimental design

No comment.

Validity of the findings

Please have a look at Major concerns: 1 to 3.

Additional comments

Breast cancer is one of the predominant types of tumors in women worldwide. Despite great progress in the early diagnosis and treatment recently, drug resistance and distal metastasis remain major causes of cancer-related mortality. Immunoglobulin superfamily member 10 (IGSF10), as a member of the immunoglobulin superfamily, is broadly expressed in both gall bladder and ovary. In this manuscript, Dr. Mengxue Wang and colleagues looked at the role of the IGSF10 in diagnosis and prognosis of breast cancer. The study’s conclusion showed that IGSF10 was expected to be used to predict the prognostic survival of breast cancer. Before further consideration of its publication, the authors must carefully revise the full text as required and provide corresponding research data as support.

Major concerns:
1. “Response to Major Q4” As can be seen from the pathological picture in Figure 1C to F, there is no typical breast cancer tissue in the micrograph!
2. “Response to Major Q5” As far as I know, the epidemic in mainland China has been well controlled in May, and I haven't heard of any laboratory that has been closed up to now.
3. “Materials & Methods - IHC” Line 120-125. “The IHC staining intensity scoring criteria were as follows: 0, none; 1, weak; 2, medium; 3, strong. The scoring criteria for the proportion of positive tumor cells were as follows: 0, < 5%; 1, 5%–25%; 2, 26%–50%; 3, 51%–75%; 4, > 75%.” The authors should explain how the immunohistochemistry score in the adjacent control tissue was calculated.

---

## Round 0.3 · accepted · Accept

Since all the critiques were adequately addressed and the manuscript revised accordingly, I am happy to accept your manuscript now.

·

Basic reporting

No comment.

Experimental design

No comment.

Validity of the findings

No comment.

Additional comments

Breast cancer is one of the predominant types of tumors in women worldwide. Despite great progress in the early diagnosis and treatment recently, drug resistance and distal metastasis remain major causes of cancer-related mortality. Immunoglobulin superfamily member 10 (IGSF10), as a member of the immunoglobulin superfamily, is broadly expressed in both gall bladder and ovary. In this manuscript, Dr. Mengxue Wang and colleagues looked at the role of the IGSF10 in diagnosis and prognosis of breast cancer. The study’s conclusion showed that IGSF10 was expected to be used to predict the prognostic survival of breast cancer. The paper is improved and most concerned raised by the reviewer have been addressed. I think it is might suitable for publication at this version of revised manuscript.